# New Mater-Bi, Biodegradable Mulching Film for Strawberry (*Fragaria* × *Ananassa* Duch.): Effects on Film Duration, Crop Yields, Qualitative, and Nutraceutical Traits of Fruits

**DOI:** 10.3390/plants11131726

**Published:** 2022-06-29

**Authors:** Luigi Morra, Maurizio Bilotto, Emiliana Mignoli, Mariarosaria Sicignano, Anna Magri, Danilo Cice, Rosaria Cozzolino, Livia Malorni, Francesco Siano, Gianluca Picariello, Sara Guerrini, Milena Petriccione

**Affiliations:** 1Council for Agricultural Research and Economics (CREA), Research Center for Cereal and Industrial Crops, 81100 Caserta, Italy; maurizio.bilotto@crea.gov.it (M.B.); emiliana.mignoli@crea.gov.it (E.M.); mariarosaria.sicignano@crea.gov.it (M.S.); 2Department of Environmental, Biological and Pharmaceutical Sciences and Technologies (DiSTABiF), University of Campania Luigi Vanvitelli, Via Vivaldi 43, 81100 Caserta, Italy; anna.magri@unicampania.it; 3Council for Agricultural Research and Economics (CREA), Research Center for Olive, Fruits and Citrus Crops, 81100 Caserta, Italy; danilo.cice@crea.gov.it (D.C.); milena.petriccione@crea.gov.it (M.P.); 4Institute of Food Science, National Research Council (CNR), Via Roma 64, 83100 Avellino, Italy; rosaria.cozzolino@isa.cnr.it (R.C.); livia.malorni@isa.cnr.it (L.M.); francesco.siano@isa.cnr.it (F.S.); gianluca.picariello@isa.cnr.it (G.P.); 5Department of Agricultural and Food Science (DISTAL), University of Bologna, Via Zamboni, 33, 40126 Bologna, Italy; sara.guerrini11@unibo.it

**Keywords:** sustainability, plastic mulch disposal, polysaccharide bio-composites, antioxidant compounds

## Abstract

In the main strawberry areas of Southern Italy, cultivation is carried out by transplanting plants on raised beds (30–40 cm from ground level), mulched with black polyethylene (PE). This technique has becoming increasingly expensive due to the growing prices of plastic mulches, the cost to remove them at the end of crop cycle, and the difficulty to dispose of black, dirty plastic films. The main objective of this research was the replacement of PE mulch with a new biodegradable mulching film Mater-Bi^®^-based (Novamont), characterized by an increased permanence in the field designed for long crop life. In 2021, two Mater-Bi-based, black, 18 μm thick mulching films were tested under tunnel: N5 as innovative film and N18 as commercial standard film. Black PE film, 50 μm thick was the control. Strawberry cultivars ‘Sabrina’ and ‘Elide’ were cropped on the three mulching films according to a split plot design with four replications. Harvests lasted from March to June 2021. Cvs Sabrina and Elide yielded around 40 t ha^−1^, while the mean effect of mulching films did not point out differences between the biodegradable mulches and PE. In 4 out of 12 harvests we analyzed samples of fruits to assess the influence of mulches on the contents of °Brix, polyphenols, antioxidant activity, ascorbic acid, flavonoids, and anthocyanins. On average, °Brix was clearly improved in fruits on PE compared to biodegradable films, while all the other qualitative traits resulted in being more dependent on the cultivars and times of sampling effects. Overall, biodegradable mulches are a viable alternative to PE mulch, and the innovative N5 film appeared promising for the enhancement of durability of soil coverage in a long-lasting cycle.

## 1. Introduction

Modern strawberry cultivation is characterized by heavy use of synthetic inputs: plastic film to cover tunnels/greenhouses, plastic mulch, plastic drip irrigation line, synthetic fertilizers, and pesticides. However, during the last 20 years, strawberry conventional cultivation has become more sustainable from an environmental and economic point of view thank to innovations like soil solarization as alternative to chemical fumigation [1] or the introduction of flowers pollinators joined to the biological control of insect pests, which has reduced the use of insecticides harmful to pollinators [2]. The massive use of plastic films for mulching represents another front for improving strawberry production environmental sustainability [3]. The environmental impacts associated with the use and disposal of low-density polyethylene plastic mulch films are of particular concern [3]. Disposal options of PE mulches include recycling, incineration while on farm burning, or onsite incorporation in the soil are forbidden practices which are still employed. However, recycling is becoming increasingly difficult because facilities tend to not accept black PE mulch that is contaminated with soil and agrochemicals [4,5]. Furthermore, plastic mulch requires expensive labor for removal and disposal. Micro or nanoparticles generated by non-biodegradable materials, may persist in soil, and affect microbial activity, physical soil properties, and nutrient availability [6]. Goldberger et al. [7] surveyed strawberry growers in California, the Pacific Northwest, and the Mid-Atlantic States of USA with the aim to explore the opinions related to the use of PE and biodegradable mulches. Survey results indicated that the adoption of biodegradable mulch may be more likely if products are compatible with local production practices such as organic production that bans their use, soil fumigation that damages them, or matted rows systems. Besides, alternative mulches should also not harm soil health and must be affordable. As demonstrated in the review of Menossi et al. [5], studies evaluating polysaccharide-based multiphase materials for effectiveness of mechanical barrier as well as biodegradability properties are growing. One possible alternative to low-density polyethylene (LDPE) mulch is given by the certified in soil biodegradable mulch films, such as those produced by Novamont and commercially available under the trade name Mater-Bi^®^. Mater-Bi is a biodegradable thermoplastic material made with starches and a biodegradable copolyester based on proprietary technology. 

Several agronomic trials demonstrated the efficacy of thermoplastic starch-based mulches for strawberry. Bilck et al. [8] tested a material based on native starch from cassava in Brazil to produce thermoplastic starch joint to polybutylene adipate-co-terephtalate (PBAT) as a biodegradable polymer, whereas DeVetter et al. [9] in the USA tested a 20 μm thick film based on Mater-Bi in open field; Costa et al. [10] tested, in Portugal, five formulations with Mater-Bi, grade NF in open field, and greenhouse conditions (EU financed project AGROBIOFILM, http://www.agrobiofilm.eu, accessed on 14 June 2022). In Italy, Morra et al. [11] tested commercial Mater-Bi, grade EF04P0, 18 μm thick, on strawberries grown in tunnels on raised beds. All these trials were intended to evaluate the performance of biodegradable films on yield and quality of strawberry, weed control, and biodegradability in soil. It is important to underline that in 2018 the European Committee for Standardization (CEN) published a standard aimed to define the end of life, impact, and performance feature of biodegradable in soil mulch film: EN 17033:2018“Plastics—Biodegradable mulch films for use in agriculture and horticulture—Requirements and test methods” [12]. The standard was adopted as reference in the European countries and biodegradable films in soil mulch are now tested according to this standard by Din Certco.

Strawberry is the most demanded berry as a fresh fruit, as well as in processed and derived food products [13]. The fruits are rich in antioxidant phytochemicals that are able to prevent the onset of chronic and degenerative diseases and maintain a state of wellbeing in humans [14]. Although many studies have been conducted to evaluate the agronomical performance of biodegradable mulch films, few studies addressed the effects on fruit qualitative and nutraceutical traits in strawberry. In Italy, according to CREA [15], total surface and production of strawberry under tunnel-greenhouses in 2020 were 2784 ha and 92,914 t (provisional data). In 2021, strawberry cultivation has experienced an important increase in terms of cultivated areas thanks to the investments that have mainly involved Basilicata (+20%), Campania (+6%), and Sicily (+4%). Southern Italy alone represents the cradle of cultivation of strawberries, harvesting 66% of production. 

In this paper the results of research carried out in 2020–2021 to assess a new Mater-Bi film, grade EF08P0 denominated N5 and optimized for longer crop cycles such as strawberries, are presented. The innovative mulching film was compared to a commercial Mater-Bi mulch, grade EF04P0 (N18), and a conventional non-biodegradable LDPE film. The objective was to evaluate, in real farm conditions, the behavior of the new improved biodegradable mulch film for the long-lasting cultivation of strawberry on high raised beds (typical of the Southern Italy). To obtain this objective, we tried to evaluate three hypotheses: improved mechanical strength and breaks of N5 mulch in comparison to commercial film N18; marketable yields of two strawberry varieties compared to yields of LDPE; improved total soluble contents and bioactive compounds in the fruit on biodegradable mulch films with respect to LDPE for four harvesting dates. 

## 2. Results and Discussion

### 2.1. Mechanical Strength of Mulching Films and Yields of Strawberry

Biodegradable mulch films were laid mechanically on the soil surface using the same equipment normally employed for conventional LDPE, as recommended by the European standard EN 13655; the mulches displayed sufficient mechanical properties and strength. After 50 days from mulch installation, 40% of the whole surface covered with N18 mulching film started showing early rips and holes along the side of the raised beds and at their bottom where soil covers and anchors the biodegradable film (Figure 1A). At the same time, N5 film showed rips only on 30% of the bottom area of the raised beds. Figure 1B,C shows the condition of N5 and N18 films, respectively, after 145 days from transplant, which corresponds to the ripening of the first fruits. N18 was detached from the bottom on 60% of raised beds while N5 mulch pointed out less widespread lesions, probably due to slower degradation.

Nevertheless, in spring, biodegradable mulching films enhanced the plant canopy as well as LDPE mulch so that coverage of raised bed top offset ongoing mulching films biodegradation. Figure 1D shows the full development of strawberry crop at the end of harvesting season after 231 days from the start of trial; timing of phenological stages of strawberry was not affected negatively by biodegradable mulching films in comparison to the standard LDPE mulching. In 2015, Morra et al. [11] carried out a trial on strawberry with Mater-Bi-based biodegradable mulching film, grade EF04P0 corresponding to N18; they found that, contrary to our actual results, the mulch remained anchored to bed bottoms after a similar long-lasting cycle. Andrade et al. [16] reported that coextruded white on black Mater-Bi films, 20 and 25 microns thick, covered a soil mulching percentage at the end of strawberry cycle of 72% and 83%, respectively.

Cultivars Sabrina and Elide produced similar marketable yields both on a per plant and per hectare basis (Table 1). Fruits categorized as ‘Extra’ accounted for 2/3 of total yield, with a fruit mean weight over 29 g, while fruits categorized as ‘First’ had a mean weight of 16–17 g. Unmarketable yield was significantly greater for Sabrina than Elide. Plants grown with N18 mulch produced greater ‘Extra’ fruit mean weight compared to plants grown with PE mulch (Table 1). Other measures of fruit yield (marketable yield per plant, not marketable yield per plant, and total marketable yield) did not differ among mulch types (Table 1). Mulch × Cultivar interaction was only significant in the case of mean fruit weight of the two commercial categories. Our results agree with the findings of [9] and [10], while Andrade et al. [16] reported a 20% mean loss of crop yield compared to PE film, suggesting that lowest soil temperature under PE in the initial summer stage of crop favored a better crop development. Giordano et al. [17] also reported the lowest yields with starch-based biodegradable films, black colored, 25 μm thick, compared to PE, 35 μm thick and Ecoflex (BASF) biodegradable, 20 μm thick film, black colored also. Authors attribute the results to the soil hydrothermal regime under biodegradable films characterized by high water vapor permeability so that soil temperature and moisture are not constant as under PE, with negative effects on flowering and fruiting of strawberry.

### 2.2. Quality of Fruits in Relation to Cultivars and Mulching Films during the Harvest Period

Three-way ANOVA underlined the significant effects of Cultivar, Mulch, Time of harvest and their first and second-order interactions for nearly all the qualitative characteristics analyzed (Table 2). Hereafter, we represented the Cultivar × Mulch × Time of Harvest interaction regarding total soluble solids (TSS), polyphenols, anthocyanins, flavonoids, and antioxidant activity. Only ascorbic acid is not represented; this parameter was significantly influenced only by the time of harvest, showing a sharp decreasing trend from 67.8 to 30.1 mg 100 g^−1^ fresh weight (f.w.) from 30 March to 1 June.

Fruits of cv Elide showed a steady TSS content in the first three sampling times and a peak in the last one (Figure 2). This behavior was common to all the mulching films, but TSS contents in the first three sampling times were significantly higher in PE (on average 8) than N5 and N18 (in the range of 6–7). Only in the last sampling date did TSS in fruits from N5 and N18 reach values like to those in PE. Fruits of cv Sabrina showed the same pattern of behavior, but TSS contents on N5 were in every sampling time significantly lower than those in fruits in N18 (Figure 2). Fruits of Sabrina ripened on PE and reached their maximum values 10.5 °Brix in advance on 17 May with respect to biodegradable films that produced fruits with TSS 9–10 °Brix in the last sampling against 8 °Brix on PE. 

In the two strawberry cultivars, PE mulching film induced in each time of sampling, except 1 June, significantly higher content of TSS compared to biodegradable mulches; TSS includes soluble sugars and other compounds of plant metabolism such as vitamins, acids, aminoacids, and pectins [18]. Values measured in our work resulted as higher as the ones found in Mendez da Silva et al. [19] and Cervantes et al. [20]. In other studies, [21,22], a great seasonal variation for TSS content was confirmed, with higher values in late harvest in several strawberry cultivars. The benefits of PE mulching film on crops physiological processes are well-documented in literature and include earlier ripening [4], with higher TSS content in strawberry fruits. DeVetter et al. [9], in Western Washington, found no differences in the levels of TSS, pH, and Citric acid measured in fruits of two strawberry cultivars grown on Mater-Bi, LDPE, and paper-based mulching films. Giordano et al. [17] also did not report significant differences in TSS measured in three harvest times. 

Mean polyphenol content was 384 and 239 mg GAE 100 g^−1^ f.w. in Elide and Sabrina, respectively (Figure 3). In other studies, the polyphenols content of cv Sabrina widely ranged from 54 to 400 mg GAE 100 g^−1^ f.w. [16,20]. Morra et al. [11], in a preceding trial with cv Sabrina grown on commercial Mater-Bi based film, grade EF04P0, measured concentrations ranging between 80–110 mg GAE 100 g^−1^ f.w.: these values represent less than half of those measured in the present research. This wide variability demonstrated the involvement of polyphenols in the plant response to multiple and simultaneous environmental changes [23,24]. In Figure 3, it is evident that the polyphenol content was noticeably influenced by cultivars and time of sampling, as demonstrated in strawberry by [25]. However, looking at values in the four different harvest times in the two cultivars, total polyphenol contents on the biodegradable films were like to PE, apart from the highest peaks measured in Elide on N18 and PE in the harvest of 17 May.

As previously stated by Wang and Zheng [26] and successively found by Fan et al. [27] in Canada, phenolic compounds increased in fruits grown on row covered by plastic film with respect to those one grown on traditional matted row systems due to higher soil temperature. On this basis, it is presumable that under biodegradable films the thermic regime was not different to than under PE, and therefore total polyphenols content in our trial did not change in relation to the mulching material [28,29,30]. 

The average anthocyanins content was lower in cv Sabrina than in cv Elide; cv Sabrina showed values comparable to other studies [11,28,29], but lower than the ones reported by [20]. Differently from Morra et al. [11], in this experiment we did not detect the same trend with higher anthocyanin content on biodegradable films in three times of sampling in two different farms. As shown in Figure 4, statistical differences were only found on 17 May when higher peaks of contents were measured in fruits of Elide grown on N5 and N18 and in fruits of Sabrina grown on N18 with respect to those ones of PE mulch at the same time of sampling. Our results, with regard to the last times of harvest when a decrease was measured in all treatments, are in agreement with [31], which report the degradation of anthocyanin content in fruits at the end of the season due to both photoinhibition and the highest photoperiod.

Flavonoid content in fruits of Sabrina followed the same pattern during the four times of sampling regarding the three mulching films (Figure 5): a steady content in the first three sampling times was detected, ranging 40–60 mg CAE 100 g^−1^ f.w., with an increase to 80–100 mg CAE, significantly more than in preceding time points on 1 June, irrespective of mulch films. Flavonoids in fruits of Elide followed the same pattern described in Sabrina only on the N18 film, while on N5 they kept a steady content of 70–80 mg CAE in all sampling times except from a lower peak on 17 May. Only on PE film was the content of flavonoids significantly higher in the first sampling time (92 mg CAE) compared to the following times when it did not exceed 60 mg CAE. Overall, the effect of the mulching film on the content of flavonoids in strawberry fruits appeared to be weak while prevailing the effect of cultivar and time of harvest. 

It is well known that colored plastic mulches represent effective tools to modify the growth environment of strawberry due to their thermal and light properties. Colored plastic mulches could alter field microclimate and affect plant surrounding light, soil temperature, and air humidity conditions. Black plastic among colored plastics has been widely used. Recent investigations show that other colored plastic mulches, especially red color, are more favorable in enhancing fruit quality and increasing fruit antioxidants such as anthocyanins and flavonoids [32,33]. Shiukhy et al. [32] found higher content of anthocyanins and flavonoids in fruits harvested on red colored plastic followed by black plastic in comparison to bare soil. On the contrary, Lalk et al. [33] did not find modifications in TSS, total polyphenols, and anthocyanins in fruits grown on black or red plastic mulch under high-tunnel in USA. In our experiment, all mulching film were black colored, so it is plausible that the plant surrounding environment was modified in a similar way by biodegradable and plastic mulches. This could explain why the flavonoids, anthocyanins, and total polyphenols appeared to vary in dependance on genotype or time of harvest rather than on kind of mulch.

In both the strawberry cultivars, antioxidant activity showed a steady trend in the first three sampling times, then decreased in the last one (Figure 6). This value ranged, overall, between 5–7 μmol Trolox equivalent g^−1^ f.w. This decreasing trend is common to Elide and Sabrina, irrespective of the mulching films. The significant lower values on 1 June were on N18 and PE films for Elide and on PE for Sabrina.

As shown in Table 3, antioxidant activity was directly correlated to anthocyanins in fruits coming from different mulching films, while TSS (°Brix) was always negatively correlated to ascorbic acids. On LDPE, however, the correlation was lower and less significant than on biodegradable films (0.46 vs. 0.76). A significant positive correlation was found between polyphenols and flavonoids in fruits harvested on the two biodegradable mulching films. Besides, on N5 film a negative correlation of 0.41 was recorded between TSS and anthocyanins, while on N18 the negative correlation of 0.45 appears between TSS and antioxidant activity. Only fruits ripened on PE mulch showed positive correlation between TSS and flavonoids, while negative correlations were detected between flavonoids and anthocyanins (0.53), polyphenols, and ascorbic acid (0.49).

### 2.3. Phenolic Compounds in Elide and Sabrina cvs on Different Mulches in the Last Harvest

Polyphenol constituents of Sabrina and Elide fruits harvested on three different mulching films (PE, N5, and N18) at the last harvest on 1 June were separated, assigned, and semi-quantified by HPLC-DAD [34]. Typical HPLC chromatograms of phenolic compounds for fruits of the two cultivars cultivated on different mulching films are shown in Figure 7, while semi-quantitative data are reported in Table 4 for cv Sabrina and in Table 5 for cv Elide. By a qualitative standpoint, the profile of phenolic compounds was conserved between the two cultivars (Figure 7).

The quantitative distribution of individual components varied depending on either the strawberry cultivar or the mulching materials. The predominant anthocyanin was in all cases pelargonidin 3-O-glucoside (P3), which was higher in Sabrina than in Elide (Table 4 and Table 5). The content of P3 in Sabrina was significantly affected by the mulching type and the highest and lowest levels were observed with the N5 and N18 mulching films, respectively (Table 4). The anthocyanin content in Elide seemed to be less affected by the biodegradable mulching type, so that P3 was unmodified between N5 and N18 films but significantly higher than PE (Table 5). The most represented phenolic compound in Sabrina was *p*-coumaryl hexoside, which is a conjugated monolignol (P1) (Table 4). In Sabrina, P1 exhibited the highest concentration with PE and the lowest one with N5 mulch (Table 4). Interestingly, the increased production of P1 corresponds to the decline in flavonoids, since *p*-coumaroyl-CoA is a common precursor to the metabolic routes leading to either flavonoid or monolignol biosynthetic pathways. The downregulation of the flavonoid pathway corresponds to increased levels of monolignols deriving from the conversion of *p*-coumaroyl-CoA [35]. In fact, Sabrina cropped on N5 mulch had the highest levels of flavonoids, such as quercetin-3-O-glucuronide (P7) and kaempferol-3-O-glucuronide (P8) (Table 4). The balance between monolignol and flavonoids was less evident for Elide, which in absolute terms contained much lower amounts of phenolic compounds than Sabrina (Table 4 and Table 5). Physiologically, the switch from the monolignol to the flavonoid biosynthetic pathways occurs during the late stages of ripening. Therefore, discrepant quantitative balances of P1 vs. flavonoids could be the results of a different ripening degree. In other terms, the mulching films may have had an effect on the ripening time of strawberries, being equal in all other conditions. This finding is in agreement with the results observed for the TSS parameter; indeed, on N5 and N18 in the first three times of sampling, TSS content was lower than PE (Figure 2). According to this evidence, in-depth studies are needed to elucidate physiological and genetic mechanisms involved in the synthesis of polyphenols on different mulching films.

## 3. Materials and Methods

### 3.1. Location and Experimental Design

The experiment was hosted in the private farm of Mr. S. Calviati, located in Falciano del Massico (Caserta) 41°140′ N, 13°944′ E. The trial was carried out under a multi-tunnels structure with a semicircular steel cross section, height 2.5 m, width 5.3 m, length 38 m. Tunnels were covered by clear ethylene-vinil acetate plastic film, 18 mm thick. Soil presented a loam texture (USDA) with clay 10.1%, silt 52.6%, sand 37.3%, organic carbon 12.7 g kg^−1^ (=2.19% organic matter), total N 1.3 g kg^−1^, C/N 9.8, pH 7.8, electrical conductivity 286 dS m^−1^, available *p* 99 mg kg^−1^. 

Two biodegradable mulching films were tested in comparison to the standard black Low-density Polyethylene (LDPE) mulching film, 50 microns thick: (a) the innovative N5 (Mater-Bi, Novamont), grade EF08P0, black colored, 18 microns thick, 1.6 m width; (b) the commercial standard N18 (grade EF04P0 Mater-Bi), black colored, 18 microns thick, 1.6 m width. The N5 film differed from N18 due to a shelf life optimized for a longer crop cycle. It was previously and successfully tested in paddy field conditions under high hydrolytic stress.

Two strawberry cultivars, namely Elide (CIV), day-neutral with low chilling requirement, and Sabrina (Planasa), short-day, were cropped on each kind of mulching film. Therefore, the experimental design was a split plot where the factor ‘mulching film’ was allocated in the main plots and the factor ‘cultivar’ was allocated in the sub-plots. The combination of all levels of two factors gave six treatments replicated 4 times. In practice, a main plot was constituted by a raised bed as long as the length of a tunnel (38.5 m), mulched with the tested films and divided in two sub-plots hosting the two cultivars. Each raised bed had isosceles trapezoidal cross section, wide 0.8 m at bottom, 0.6 m at top, and high 0.35 m at both sides. Each raised bed was 0.4 m apart from the other. The whole area devoted to the experiment was 1200 m^2^. Inside each experimental plot, an area was delimited containing 10 plants in which mature fruits were harvested to assess yields and quality of the production.

### 3.2. Agronomic Trial Management

Soil was fertilized with 80 Mg ha^−1^ of cow manure, then soil solarization was applied in July–August 2020 to maximize the toxic effects of ammonium and organic acids released from fresh manure on pathogen microflora. After solarization soil was rototilled and furtherly fertilized with 250 kg ha^−1^ of Maxilife (Intertech), a biofertilizer based on a consortium of Trichoderma spp., mycorrhizal fungi, and plant growth promoting microorganisms; 400 kg ha^−1^ of Pheoscor (Timac) (=64, 32, 8, and 88 kg ha^−1^ of phosphorus (P_2_O_5_), calcium (CaO), magnesium (MgO), and sulfur (SO_3_), respectively). 

Bed former shaper and mulching of raised beds were mechanically executed on 10 October 2020, without technical difficulties for the biodegradable films. Fresh strawberry plants (bare roots) of Sabrina and Elide were transplanted on 10 October 2020, in double rows per each raised bed according to a planting distance of 30 cm between rows and 25 cm between plants on the row. The distance between the center of two adjacent raised beds was 1.2 m; therefore, plant density was 6.6 plants m^−2^. Plant nutrition management was carried out in fertigation via driplines under mulch, supplying only Fe and Mg every two weeks alternating with Ca. Biological control of spider mite and thrips was obtained by launches of *Phytoseiulus persimilis* and *Orius laevigatus,* respectively.

### 3.3. Yields Assessment and Fruit Sampling

Harvestings started on 11 March 2021. We collected marketable and not marketable fruits weekly in delimited areas of 10 plants in each replicate. Fruits were culled in two commercial grades, extra and I category, numbered and weighted; non-marketable fruits were only weighted. Fruits of Extra category had a mean weight higher than 25 g, regular shape and color, and lack of faults, while fruits of I category had a mean fruit weight higher than 15 g, regular shape, and color. Harvestings ended on 1 June after 12 passages. In 4 out of 12 harvestings, precisely on 30 March, 4 and 17 May, 1 June, five completely red fruits were sampled from the collected product in each experimental plot to undertake laboratory analysis of the following quality and nutraceutical parameters: TTS (°Brix), ascorbic acid, total polyphenols, anthocyanins, flavonoids, and antioxidant activity. Only in the last harvest were fruits also analyzed to determine the semi-quantitative determination of polyphenols.

### 3.4. Qualitative and Nutraceutical Traits Analysis

TTS (°Brix) was determined in strawberries juice using a digital refractometer (DBR35; Sinergica Soluzioni, Pescara, Italy). Bioactive compounds of strawberries fruit (1:10 *w*/*v*) were extracted in methanol/water (80:20 *v*/*v*). The total polyphenol content (TP) was quantified by the Folin–Ciocalteu method using gallic acid as standard and results were expressed as g of gallic acid equivalents 100 g^−1^ FW (fresh weight) as described by [36]. The total flavonoid content was determined using the aluminum chloride colorimetric method and expressed as g of catechin equivalents 100 g^−1^ FW, as reported by [37]. The total monomeric anthocyanins were detected by the pH differential method in agreement with [38] and expressed as g of cyanidin- 3-glucoside equivalents 100 g^−1^ FW. Free radical scavenging activity was quantified by using DPPH assay according to the method described by [39] and the results were expressed as nmol Trolox equivalents g^−1^ FW. Ascorbic acid content was determined following the method described by [40] and the results were expressed as g ascorbic acid kg^−1^ FW (AA).

### 3.5. Reversed Phase-High Performance Liquid Chromatographic-Diode Array Detector (RP-HPLC-DAD)—Semi-Quantitative Determination of Polyphenols

Strawberry methanolic (80%, *v*/*v*) extracts were ten-fold diluted with aqueous 0.1% (*v*/*v*) trifluoroacetic acid (TFA) and analyzed as previously described [27]. Briefly, 100 µL of the diluted sample was separated using a modular HP 1100 chromatographer (Agilent Technologies, Paolo Alto, CA, USA) equipped with a 250 × 2.0 mm i.d. C18 reversed-phase column, 4 mm particle diameter (Jupiter Phenomenex, Torrance, CA, USA) held at 37 °C in a thermostatic oven. HPLC runs were performed at a constant flow rate of 0.2 mL/min applying the following gradient of solvent B: isocratic elution at 5% B for 5 min, 5–60% linear gradient of B for 5–65 min, and 60–100% B at 65–70 min. Eluent A and B were 0.1% TFA in HPLC-grade water and 0.1% TFA in acetonitrile, respectively. Effluents were monitored at wavelengths λ = 520, 360, 320, and 280 nm using a diode array detector (DAD), also acquiring an UV-Vis spectrum every second in the 200–700 nm range. Phenolic compounds were identified through the convergent indications from UV-Vis spectra, liquid chromatography coupled with high resolution tandem mass spectrometry (LC-MS/MS) [27] and comparison with literature data, and confirmed through the parallel analysis of self-prepared multicomponent standards. 

Flavonoids were semi-quantified at 360 nm using an external calibration curve built with standard rutin, while antocyanins and coumaryl-hexoside were semi-quantified based on calibration curves built with cyanidin-3-O-glucoside and *p*-coumaric acid with absorbance at 520 and 320 nm, respectively. Data were processed using the ChemStation software (version A.10) purchased with the chromatograph. Each sample was analyzed in triplicate and peak area values were averaged.

### 3.6. Statistical Analysis 

Shapiro–Wilk procedure was performed to verify if the data had a normal distribution, while Levene’s Test was conducted to verify the homogeneity of variances. Successively, data related to yields were analyzed by factorial ANOVA with sources of variability attributed to ‘Mulching film’, ‘Cultivar’, and their interaction. Fruit quality traits were analyzed adding to the model adopted for strawberry yields, the factor ‘Time of harvest’, and all possible interactions of the three factors. When the effect of a source of variability was significant, means were separated by Tukey HSD test (*p* = 0.05). Some of the most explaining significant interactions of second order related to fruit quality were represented by bar charts. Pearson’s correlation among the six fruit qualitative and nutraceutical traits were calculated per each kind of mulching material (*n* = 24). Statistical analyses were conducted by using software JMP v 16.2 (SAS Institute, Cary, NC, USA).

## 4. Conclusions

The innovative biodegradable Mater-Bi film, N5, was tested in a strawberry cultivation over a 230-day period. The optimized shelf-life characteristics of N5 resulted as effective and improved the performance in terms of durability of soil coverage in comparison to the reference mulch film N18. Both biodegradable mulch films, however, were able to guarantee a good crop productivity: crop yields per plant were 677, 644, and 587 g on N5, N18, and PE, respectively. Thickness of the tested biodegradable mulching films was 18 microns, while in trials carried out in USA [9] and Portugal [10] it was successful with a 20 microns-thick Mater-Bi films. As a matter of fact, in the main strawberry-growing areas of southern Italy (Campania and Basilicata), the agronomical technique is highly challenging for a biodegradable mulch film due to the high soil mass contained in the volume of raised beds. For this reason, a thicker film over 20 microns would be advisable. Finally, concerning the qualitative characteristics of strawberry fruits grown on the two types of mulch films (biodegradable and non-biodegradable), our findings underlined that, with the exception of TSS, the variations of contents found in these characteristics mainly depend on the different cultivars employed and the harvest sampling time. Further studies are needed on N5 innovative mulch film for strawberries to better understand the relations between the biodegradable mulch film, the role of its thickness, the environmental parameters (soil and air temperature, soil water content, and spectral and photosynthetic characteristics), fruit quality, and yield.

## Figures and Tables

**Figure 1 plants-11-01726-f001:**
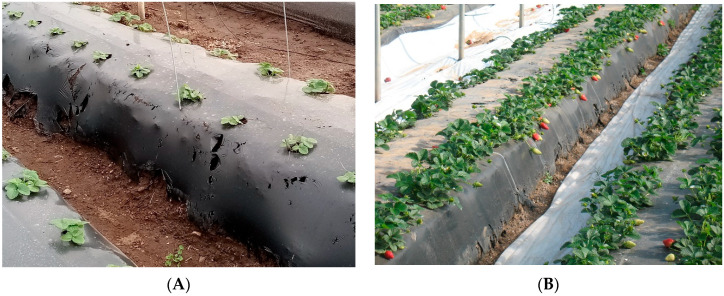
Rips and holes along the side walls and at bed bottoms for N18 mulch after 50 days after installation (**A**); Rips at the bottom of N5 mulched raised beds after 145 days from mulch installation (**B**); Rips at the bottom of N18 mulched raised beds after 145 days from mulch installation (on the left the N5 mulched raised beds) (**C**); N5 mulching film and crop development at the last harvest after 231 days mulch installation (**D**).

**Figure 2 plants-11-01726-f002:**
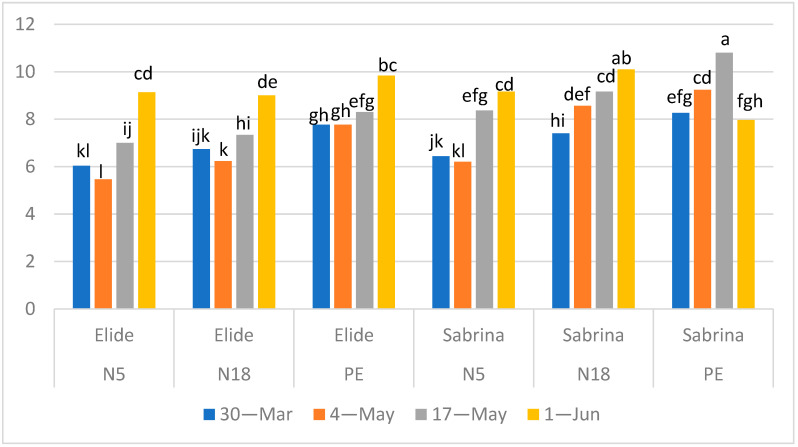
Effect of Mulch × Cultivar × Time of sampling interaction (*p* < 0.0001) in total soluble solid content (° Brix). Means followed by the same letter do not differ significantly at *p* = 0.05 (Tukey HSD test).

**Figure 3 plants-11-01726-f003:**
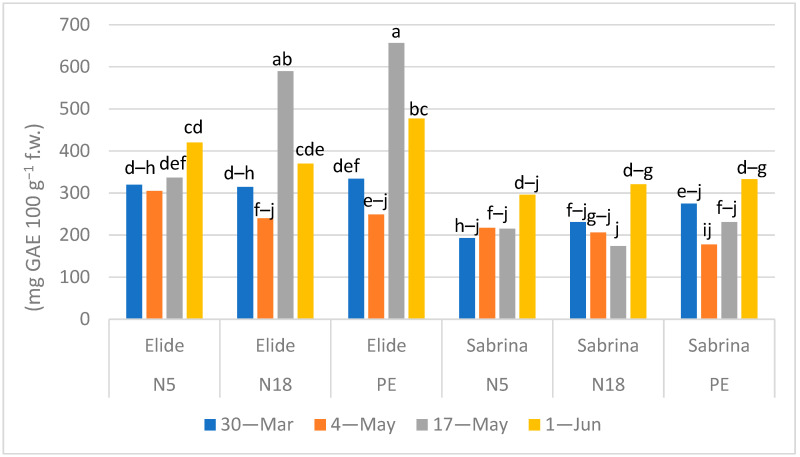
Effect of Mulch × Cultivar × Time of sampling interaction (*p* < 0.0001) in polyphenols content (mg GAE 100 g^−1^ f.w.). Means followed by the same letter do not differ significantly at *p* = 0.05 (Tukey HSD test).

**Figure 4 plants-11-01726-f004:**
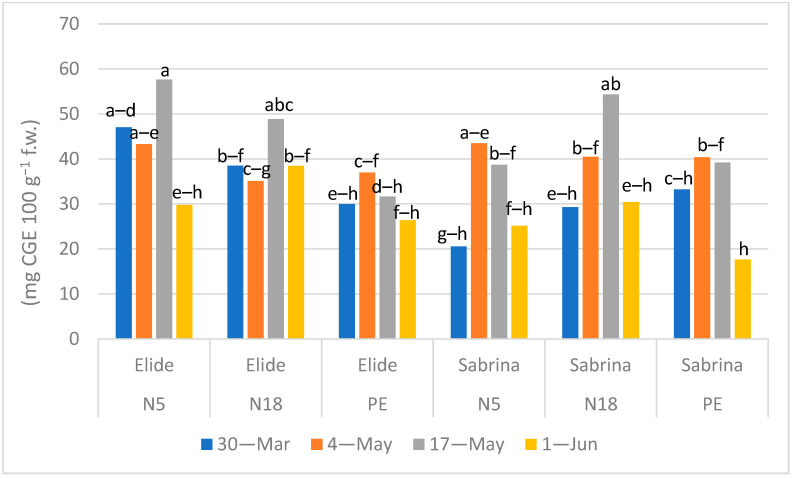
Effect of Mulch × Cultivar × Time of sampling interaction (*p* < 0.0001) in anthocyanins content (mg CGE 100 g^−1^ f.w.). Means followed by the same letter do not differ significantly at *p* = 0.05 (Tukey HSD test).

**Figure 5 plants-11-01726-f005:**
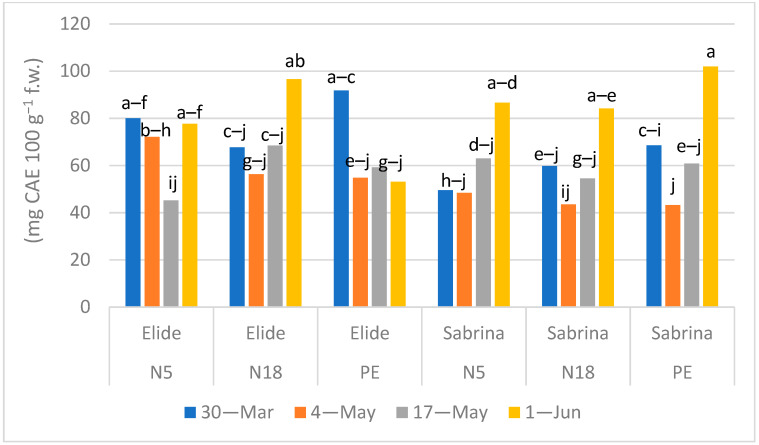
Effect of Mulch × Cultivar × Time of sampling interaction (*p* < 0.0001) in flavonoids content (mg CAE 100 g^−1^ f.w.). Means followed by the same letters do not differ significantly according to Tukey HSD Test (*p* = 0.05).

**Figure 6 plants-11-01726-f006:**
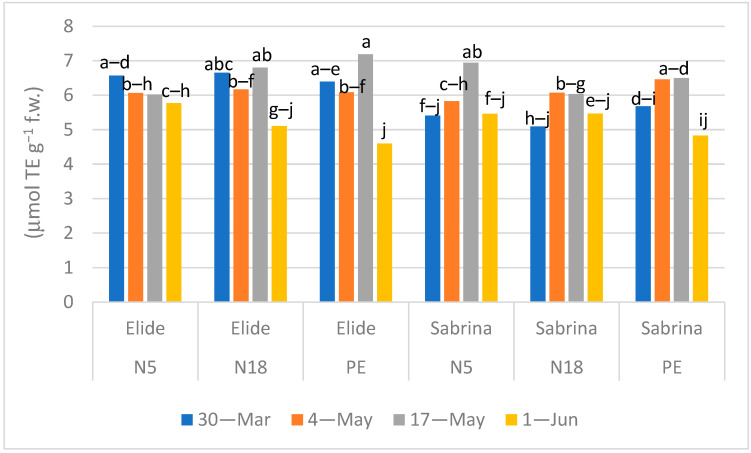
Effect of Mulch × Cultivar × Time of sampling interaction (*p* < 0.0001) in antioxidant activity. Means followed by the same letters do not differ significantly according to Tukey HSD test (*p* = 0.05).

**Figure 7 plants-11-01726-f007:**
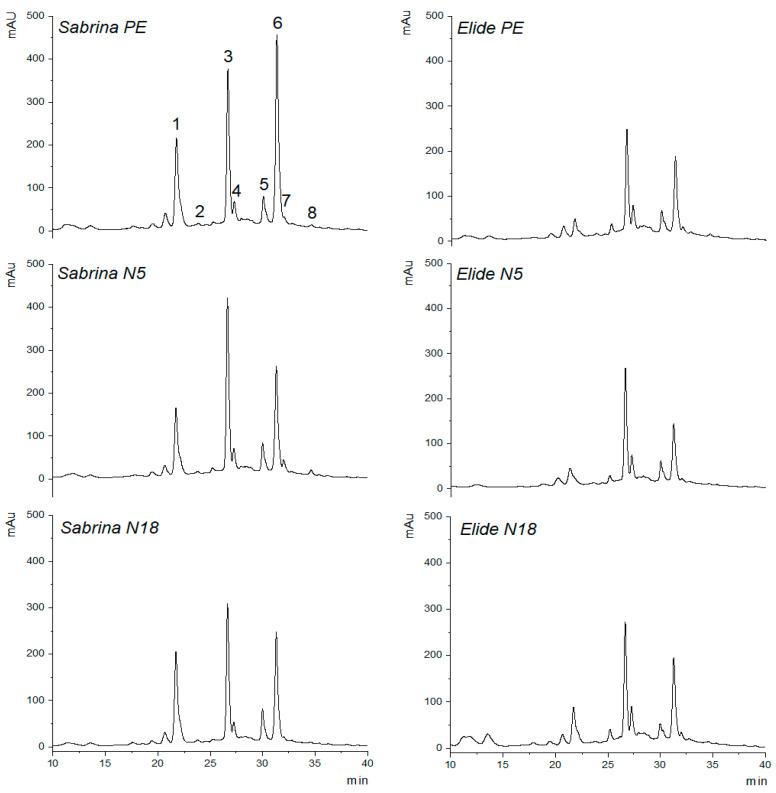
HPLC chromatograms of phenolic compounds in fruits of the cvs Sabrina and Elide cropped on PE, N5, and N18 mulching films.

**Table 1 plants-11-01726-t001:** Strawberry cultivar, mulch type, and cultivar × mulch type interaction effects on mean fruit weight, mean marketable yield per plant, mean not marketable yield per plant, and total marketable yield per hectare.

	Mean Fruit Weight (g)		Marketable Yield per Plant (g)			Non-Marketable Yield per Plant (g)	Total Marketable Yield
**Cultivar**	**Extra**	**First**	**Extra**	**First**	**Total**		**(t ha^−1^)**
Sabrina	29.4	17.1	416.2	211.3	627.5	22.6	41.4
Elide	29.6	16.5	451.3	194.3	645.6	14.7	42.0
*p*	*n.s.*	*n.s.*	*n.s.*	*n.s.*	*n.s.*	****	*n.s.*
**Mulch**							
MB N5	29.5 ab	16.3	460.2	217.1	677.3	19.6	44.7
MB N18	30.6 a	17.7	451.8	193.0	644.8	17.7	42.5
PE	28.5 b	16.3	389.3	198.4	587.7	18.6	37.9
*p*	****	*n.s.*	*n.s.*	*n.s.*	*n.s.*	*n.s.*	*n.s.*
**Mulch × Cultivar**							
*p*	****	***	*n.s.*	*n.s.*	*n.s.*	*n.s.*	*n.s.*

Means followed by the same letter do not differ significantly at *p* = 0.05 (Tukey HSD). *n.s*. not significant; * *p* < 0.05, ** *p* < 0.01.

**Table 2 plants-11-01726-t002:** Three-way ANOVA table representing the statistical significance of the effects of Cultivar, Mulch, Time of Harvest and their interactions on total soluble solids content (TSS), ascorbic acid, polyphenols, anthocyanins, flavonoids, and antioxidant activity in strawberry fruits.

	TSS (°Brix)	Ascorbic Acid	Polyphenols	Anthocyanins	Flavonoids	Antioxidant Activity
		(mg 100 g^−1^ f.w.)	(mg GAE 100g^−1^ f.w.)	(mg CGE 100 g^−1^ f.w.)	(mg CAE 100 g^−1^ f.w.)	(μmol TE g^−1^ f.w.)
**Cultivar**	***	*n.s.*	***	***	*	***
**Mulch**	***	*n.s.*	***	***	*n.s.*	*n.s.*
**Time**	***	***	***	***	***	***
**Mulch× Cv**	***	*	*	***	**	*n.s.*
**Mulch × Time**	***	**	***	***	***	***
**Cv × Time**	***	***	***	***	***	***
**Mulch× Cv × Time**	***	*n.s.*	***	***	***	***

*n.s*. not significant; * *p* < 0.05, ** *p* < 0.01; *** *p* < 0.001.

**Table 3 plants-11-01726-t003:** Significant Pearson correlation coefficients among qualitative and nutraceutical traits measured both in ‘Sabrina’ and ‘Elide’ as the mulching films changes (*n* = 24). Significance: *** significant for *p* ≤ 0.001; ** significant for *p* ≤ 0.01; * significant for *p* ≤ 0.05.

MULCH N5	°Brix	Polyphenols	Antioxidant Activity	Anthocyanins
°Brix				
Polyphenols				
Antiox. Activ.				
Anthocyanins	−0.41 *		0.56 **	
Ascorbic Ac.	−0.76 ***			
Flavonoids		0.53 **		
**MULCH N18**				
°Brix				
Polyphenols				
Antiox. Activ.	−0.45 *			
Anthocyanins			0.48 **	
Ascorbic Ac.	−0.76 ***			
Flavonoids		0.50 **		
**PE**				
°Brix				
Polyphenols				
Antiox. Activ.				
Anthocyanins			0.45 *	
Ascorbic Ac.	−0.46 *	−0.49 **		
Flavonoids	0.46 **			−0.53 **

**Table 4 plants-11-01726-t004:** Assignment and semi-quantitative determination of phenolic compounds (mg 100g^−1^ FW) measured in strawberries cv Sabrina cultivated with three different mulch types (PE, N5, and N18).

Polyphenols	Code	PE		N5		N18		*p*
p-coumaryl hexoside	P1	47.5 ± 0.4	a	34.5 ± 0.2	c	43.1 ± 0.3	b	***
cyanidin-3-O-glucoside	P2	0.4 ± 0.0	b	0.7 ± 0.0	a	0.2 ± 0.0	c	***
pelargonidin 3-O-glucoside	P3	20.7 ± 0.2	b	23.9 ± 0.2	a	17.0 ± 0.0	c	***
pelargonidin 3-rutinoside	P4	2.3 ± 0.0	a	2.2 ± 0.1	a	1.8 ± 0.0	b	***
quercetin-3-O-glucoside	P5	1.8 ± 0.1	a	1.8 ± 0.1	a	1.1 ± 0.0	b	***
kaempferol-3-O-glucoside	P6	1.2 ± 0.0	a	1.3 ± 0.1	a	0.7 ± 0.1	b	***
quercetin-3-O-glucuronide	P7	2.7 ± 0.1	b	11.2 ± 0.2	a	1.2 ± 0.1	c	***
kaempferol-3-O-glucuronide	P8	2.8 ± 0.1	b	4.9 ± 0.1	a	1.5 ± 0.0	c	***

For each parameter, the mean values followed by different letters are significantly different (*p* ≤ 0.05) according to least significant difference (LSD) test. Significance: ns = not significant; *** significant for *p* ≤ 0.001;

**Table 5 plants-11-01726-t005:** Assignment and semi-quantitative determination of phenolic compounds (mg 100g^−1^ FW) measured in strawberries cv Elide cultivated with three different mulch types (PE, N5, and N18).

Polyphenols	Code	PE		N5		N18		*p*
p-coumaryl hexoside	P1	8.5 ± 0.2	*c*	10.8 ± 0.1	*b*	17.4 ± 0.1	*a*	***
cyanidin-3-O-glucoside	P2	1.4 ± 0.0	*b*	1.1 ± 0.1	*c*	1.6 ± 0.1	*a*	***
pelargonidin 3-O-glucoside	P3	13.2 ± 0.1	*b*	14.4 ± 0.1	*a*	14.5 ± 0.3	*a*	***
pelargonidin 3-rutinoside	P4	1.7 ± 0.0	*c*	2.1 ± 0.1	*a*	2.0 ± 0.1	*b*	***
quercetin-3-O-glucoside	P5	2.5 ± 0.1	*a*	1.8 ± 0.1	*b*	2.4 ± 0.2	*a*	***
kaempferol-3-O-glucoside	P6	1.8 ± 0.0	*b*	1.3 ± 0.1	*c*	2.1 ± 0.2	*a*	***
quercetin-3-O-glucuronide	P7	4.8 ± 0.0	*b*	2.4 ± 0.2	*c*	5.5 ± 0.3	*a*	***
kaempferol-3-O-glucuronide	P8	2.3 ± 0.0	*a*	1.3 ± 0.1	*c*	1.7 ± 0.1	*b*	***

For each parameter the mean values followed by different letters are significantly different (*p* ≤ 0.05) according to least significant difference (LSD) test. Significance: ns = not significant; *** significant for *p* ≤ 0.001.

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
