# Peer review of "New Mater-Bi, Biodegradable Mulching Film for Strawberry (Fragaria × Ananassa Duch.): Effects on Film Duration, Crop Yields, Qualitative, and Nutraceutical Traits of Fruits"

_plants, 2022, doi:10.3390/plants11131726_

Round 1
Reviewer 1 Report
General comments
This is an experiment concerning the effects of a new Mater-Bi biodegradable mulching film for strawberry on film duration, crop field, qualitative and nutraceutical traits of fruits. This paper is of good value and relevance for improving the environmental sustainability of the strawberry crop. However, the manuscript could be improved. It should be written more clearly and presented in a well-structured way.
Apart from the issues shown with ‘track changes’ throughout the manuscript, there are some points that need to be addressed. See the comments.
Article
The manuscript should be rearranged as follow: introduction, materials and methods, results and discussion, and conclusions.
When citing several references within the text it should be as follow:
Example: line 63 - [4,5]
Line 169 – “… to those found by Mendes da Silva et al. (2021) [16] and Cervantes et al. (2020) [17].
Introduction
The last paragraph should move to the materials and methods section.
The authors should state purpose of experiment and hypotheses clearly.
Material and Methods
The mechanical strength of the films was assessed through visual evaluations. Was there a visual scale degradation of mulch cover?
Results and Discussion
In this section writing should be done in a clear, simple and concise manner.
Examples:
Line 143 – “As showed in Table 1, cvs. Sabrina …..” “ Cultivars Sabrina and Elide pointed out similar ……hectare (Table 1).
Line 144 – “Their fruits Extra ……” – “Extra category fruits accounted…”
Line 163 – “The results of three way ANOVA for ….” – “The effects of strawberry cultivars….”
Figure 1 – the photos do not clearly illustrate the differences between N5 and N18 BD mulch films. The photos to illustrate the degradation of N5 and N18 films should be of the same time to be comparable.
Conclusions
Clearly state the conclusions and whether the results support or reject the hypotheses of the experiment.
N5 and N18 mulching films had the same performance on fruit yield and fruit quality. Is there any advantages in using N5 instead of N18 (price cost, for example?).

Reviewer 2 Report
Congratulations for your interesting work!
The data in the manuscript is interesting and useful. There are some small editing issues that I think you might adjust, ie. in line 31 - ...black Polyethylene (PE); in line 36 - 18 μm; in line 38 - 50 μm, etc.
Author Response
We corrected the few and minor suggestions provided by reviewer 2 in his report
Reviewer 3 Report
Your presentation lacks discussion about possible mechanistic explanations for your results. Mulches were applied then strawberry attributes were measured and reported, but the discussion does not shed light on WHY the mulches performed differently. A more useful approach would have been to measure some environmental variables that could help explain differences. At least, the discussion could use some literature citations to speculate about causality underlying your observational results.
The writing also needs substantial improvement. Unfortunately, the journal does not provide a file that is easily edited. Reviewers are issued a PDF file, so I made changes as I could within this PDF (see uploaded file). I made detailed changes in the text down to line 169 - the changes are not tracked but I rewrote nearly every sentence. The rest of the writing needs substantial improvement, preferably by someone fluent in both English writing and scientific writing.
